# Analysis of Neuronal Excitability Profiles for Motor-Eloquent Brain Tumor Entities Using nTMS in 800 Patients

**DOI:** 10.3390/cancers17060935

**Published:** 2025-03-10

**Authors:** Ismael Moser, Melina Engelhardt, Ulrike Grittner, Felipe Monte Santo Regino Ferreira, Maren Denker, Jennifer Reinsch, Lisa Fischer, Tilman Link, Frank L. Heppner, David Capper, Peter Vajkoczy, Thomas Picht, Tizian Rosenstock

**Affiliations:** 1Department of Neurosurgery, Charité Universitätsmedizin Berlin, 10117 Berlin, Germany; ismael.moser@charite.de (I.M.); melina.engelhard@charite.de (M.E.); felipe.montesanto@charite.de (F.M.S.R.F.); maren.denker@charite.de (M.D.); jennifer.reinsch@charite.de (J.R.); lisa.fischer@charite.de (L.F.); peter.vajkoczy@charite.de (P.V.); thomas.picht@charite.de (T.P.); 2Einstein Center for Neurosciences, Charité Universitätsmedizin Berlin, 10117 Berlin, Germany; 3International Graduate Program Medical Neurosciences, Charité Universitätsmedizin Berlin, 10117 Berlin, Germany; 4Institute of Biometry and Clinical Epidemiology, Charité Universitätsmedizin Berlin, 10117 Berlin, Germany; ulrike.grittner@charite.de; 5Department of Neuropathology, Charité Universitätsmedizin Berlin, 10117 Berlin, Germany; frank.heppner@charite.de (F.L.H.); david.capper@charite.de (D.C.); 6Cluster of Excellence, NeuroCure, Charitéplatz 1, 10117 Berlin, Germany; 7German Center for Neurodegenerative Diseases (DZNE), 10117 Berlin, Germany; 8German Cancer Consortium (DKTK), Partner Site Berlin, German Cancer Research Center (DKFZ), 69120 Heidelberg, Germany; 9Cluster of Excellence Matters of Activity, Image Space Material, Humboldt Universität zu Berlin, 10178 Berlin, Germany; 10Berlin Institute of Health (BIH), Charité Universitätsmedizin Berlin, 10117 Berlin, Germany

**Keywords:** navigated transcranial magnetic stimulation, motor excitability, resting motor threshold, motor evoked potential, brain tumor, glioma, motor cortex, corticospinal tract, neuroplasticity

## Abstract

Navigated transcranial magnetic stimulation (nTMS) is a non-invasive diagnostic brain stimulation tool that is used to assess anatomy and functionality of the motor system. Patients with brain tumors in motor regions showed altered motor excitability in different nTMS parameters. Recent research has explored the biological tumor entity as one factor influencing this excitability. However, specific alteration patterns for different brain tumor entities are not established yet. Therefore, we analyzed the relationship between various nTMS motor parameters and different tumor entities in a large cohort of 800 brain tumor patients. We identified characteristic alterations in nTMS motor excitability profiles for certain biological tumor entities. Entity-specific brain–tumor interaction in the motor system reflects different mechanisms of plasticity. Insights into the biological behavior of different tumor entities can enhance the assessment of the surgical risk and may potentially identify approaches to induce plasticity in the motor system by nTMS-based neuromodulation.

## 1. Introduction

Navigated transcranial magnetic stimulation (nTMS) is used as a non-invasive tool for anatomical-functional mapping of the motor system. The detection of motor-evoked potentials (MEPs) in electromyography (EMG) depending on brain stimulation points leads to the creation of individual cortical motor maps. The combination of the nTMS data with diffusion tensor imaging (DTI) enables a function-based fiber tracking of the corticospinal tract (CST) [1,2]. Due to the precise neuronavigated stimulation, high accuracy compared to the gold standard of intraoperative direct cortical stimulation has been shown [3,4]. Compared to fMRI, which provides indirect functional data based on hemodynamic responses, nTMS allows for targeted stimulation of motor areas with real-time electrophysiological feedback. In comparative studies, nTMS clearly outperformed fMRI in terms of accuracy and reliability compared to DCS [5]. Additionally, the functionality of the motor system can be objectified by neurophysiological nTMS parameters such as the resting motor threshold (RMT), motor area size, MEP amplitude and MEP latency. Thus, nTMS motor mapping has proven to be useful for patients with motor-eloquent brain tumors, as it enables preoperative assessment of integrity and functionality in the motor system and optimizes planning and execution of surgical resection [6,7,8,9]. Therefore, maximal tumor resection can be achieved while reducing the risk of new or worsened motor deficits [10,11,12,13,14]. NTMS-based prognostic risk stratification models have been developed and are incorporated into clinical neurosurgical decision making [15,16,17,18]. Moreover, nTMS has already been used to assess tumor- or resection-induced neuroplastic reorganization of the brain [19,20,21,22,23].

Recent studies have begun to explore how different tumor entities affect motor excitability measured by nTMS, revealing that biomolecular differences significantly influence tumor growth behavior and the plasticity of eloquent brain regions [24]. Thus, clinical manifestation, therapeutical options and a patient’s prognosis are influenced [25]. Research on motor excitability in brain metastases and benign tumors has been scarce [26,27,28,29], while analyses of glioma subtypes have been more frequently conducted [22,28,30,31,32,33]. Lavrador et al. found that pathological excitability levels (measured by interhemispheric RMT ratios) were more frequently observed in high-grade gliomas [31,32,33]. However, characteristic and comprehensive motor excitability patterns for certain brain tumor entities have not been established or validated yet, and possible underlying mechanisms are not sufficiently understood. This may also be attributed to significant inter-individual variability of nTMS parameters. Research involving both healthy individuals [34,35,36] and tumor patients [27,37,38,39] has explored various subject- or tumor-specific factors that contribute to this variability. For example, while larger tumor volumes may lead to reduced excitability due to structural infiltration or compression of the motor cortex, peritumoral edema can cause cortical disinhibition through inflammation, potentially resulting in increased excitability. Since tumor entities also differ in such specific factors, relevant confounders must be considered.

The aim of this study was to systematically investigate nTMS motor excitability profiles across various brain tumor entities, with a particular focus on glioma subtypes, in a large cohort of 800 patients with motor-eloquent brain tumors. Specifically, this study addresses the following research questions: (1) How do motor excitability profiles differ among various brain tumor entities, particularly gliomas, metastases, and benign lesions? (2) How do glioma subtypes influence motor excitability? (3) What neurophysiological alterations can be observed in patients with paresis? By analyzing a large patient cohort using a confounder-adjusted statistical approach, we aim to provide new insights into tumor-entity-specific alterations of motor excitability and their potential impact on preoperative risk stratification and neurosurgical decision-making.

## 2. Materials and Methods

### 2.1. Study Design and Population

This observational study was conducted as a single-center retrospective analysis at a large university hospital between October 2007 and May 2021. The study protocol was approved by the local ethics committee. Due to the retrospective nature of the study, additional informed consent was waived. All procedures were conducted in accordance with the ethical standards of the Declaration of Helsinki and the STROBE guidelines for transparent and structured reporting of observational studies. Patients were included if they met the following criteria: (1) age ≥ 18 years, (2) presence of a motor-eloquent brain tumor, defined as a lesion in proximity to, compressing, or infiltrating the primary motor cortex (M1) or corticospinal tract (CST), and (3) completion of nTMS motor mapping as part of the preoperative diagnostic routine. Exclusion criteria included the presence of intracranial implants, missing MRI data due to emergency surgery or non-MRI-compatible devices (e.g., pacemakers) or incomplete nTMS recordings. To maintain high methodological quality, predefined nTMS parameters were analyzed as part of a standardized nTMS mapping protocol. Moreover, uniform data acquisition of preselected patient- and tumor-specific variables was performed. The study population was documented systemically and anonymously in the Research Electronic Data Capture (REDCap) database. A detailed flowchart of the patient selection process is provided in Figure 1.

### 2.2. Patient and Clinical Data

Age and sex of the patients were documented. Handedness was assessed using the Edinburgh Handedness Inventory [40]. The intake of antiepileptic drugs (AEDs) because of suspected or known epileptic seizures was assessed. Because of the high coincidence between seizures and AED medication, we only evaluated the presence of AED medication for further analysis. Motor strength of the upper extremity was determined according to British Medical Research Council (BMRC). Motor deficit was defined as BMRC ≤ 4/5, meaning reduced muscle strength but retained movement. As motor assessment relies on clinical examination, inter-examiner variability and patient-specific factors (e.g., fatigue, medication effects) may influence results. To mitigate bias, assessments were performed by experienced neurosurgeons following standardized protocols.

### 2.3. Magnetic Resonance Imaging

All patients received an MRI using a 1.5 or 3 T unit (GE Healthcare) with an 8-channel head coil. Motor-eloquent tumor location was verified by an interdisciplinary team of neurosurgeons and neuroradiologists. The tumor hemisphere was classified as dominant or non-dominant according to handedness. Presence of more than one tumor focus was considered as multifocal. However, all further tumor data refer only to the motor-eloquent focus. A contrast-enhanced 3D gradient-echo sequence (MP-Rage, isotropic voxel size 1 mm) was used for nTMS mapping and for volumetry of contrast-enhancing tumors. The fluid-attenuated inversion recovery (FLAIR) sequence was utilized to measure volume of non-contrast-enhancing tumors and the T2 sequence for peritumoral edema. Volume measurements were performed with the planning software Elements 2.0 (Brainlab AG, Munich, Germany). Additionally, a DTI sequence was recorded, which was used for nTMS-based fiber tracking.

### 2.4. NTMS Mapping and Data Processing

All patients underwent motor mapping of the hand by biphasic magnetic stimulation through an eight-shaped stimulation coil using NBS 5.1 or Nexstim Eximia (Nexstim, Helsinki, Finland) following a standardized protocol [41]. The EMG activity of the first dorsal interosseus (FDI) muscle was recorded using Neuroline 720 electrodes (Ambu, Copenhagen, Denmark). The quality of resting EMG was checked during measurement sequences to eliminate false-positive MEPs. The motor excitability was assessed by four nTMS parameters (RMT, area, amplitude and latency) for both hemispheres (Figure 2). The extent of nTMS mapping varied based on factors such as patient compliance, technical feasibility and time constraints, resulting in incomplete measurements of nTMS parameters for some cases.

#### 2.4.1. RMT

The RMT was determined either manually using the lowest stimulation intensity to generate at least 5 MEPs (≥50 μV) in 10 stimulations or automatically using a software-integrated algorithm, which has already been published elsewhere [42]. RMT was measured as the electric field strength (V/m) on the anatomical individual cortex surface. The interhemispheric RMT ratio was calculated (RMT_Ratio_ = RMT_Sick_:RMT_Healthy_ × 100) and documented as absolute RMT_Ratio (%)_ as well as binary classified in RMT_Ratio (Pathologic)_ (<90% or >110%) or RMT_Ratio (Physiologic)_ (90–110%).

#### 2.4.2. Motor Area

The cortical motor area of the FDI was determined by mapping with a stimulation intensity of 105% of the RMT, so that MEP-positive points were surrounded by MEP-negative points. The area (mm^2^) of MEP-positive stimulation points was calculated using the Convex Hull method in Matlab R2021a (Mathworks Inc., Natick, MA, USA) [43].

#### 2.4.3. Amplitude and Latency

Peak-to-peak amplitude (μV) and latency time from nTMS stimulation until MEP onset (ms) was determined by calculating the mean of the five largest MEPs in the cortical motor area stimulated with an intensity of 105% of the RMT.

#### 2.4.4. Motor-Eloquent Tumor Location

The spatial relation between tumor and anatomical-functional motor system was evaluated based on the nTMS-based risk stratification, evaluating motor cortex (M1) infiltration and tumor-tract-distance (TTD) [44].

### 2.5. Neuropathological Diagnosis

Diagnosis of the tumor entity was performed by the local neuropathological institute. Neuropathological diagnosis of gliomas was performed based on the current WHO classification, which defines glioma entities based on histopathological and molecular criteria. [45]. IDH status and 1p19q codeletion were analyzed, as these biomarkers are crucial for distinguishing between glioblastomas (IDH-wildtype), astrocytomas (IDH-mutated) and oligodendrogliomas (IDH-mutated and 1p19q-codeleted). IDH mutation status was determined via immunohistochemistry and confirmed by DNA sequencing in cases where results were inconclusive. The presence of 1p19q codeletion was assessed using fluorescence in situ hybridization (FISH) or polymerase chain reaction (PCR)-based methods. The WHO grade of gliomas was assigned according to histopathological features such as mitotic activity, necrosis, and microvascular proliferation. While glioblastomas are defined as WHO grade 4, astrocytomas can be classified as WHO grade 2 to 4, and oligodendrogliomas as WHO grade 2 to 3.

For brain metastases, tumor origin was confirmed through immunohistochemical markers in combination with matching clinical-radiological findings. Non-malignant lesions, such as meningiomas and vascular malformations, gliosis/reactive brain, encephalitis and WHO grade 1 neuronal/neuroglial brain tumors were grouped as benign entities to create a sufficient sample size for further analysis. Unclear/unclassifiable entities and also lymphomas were excluded from the final analysis due to low case numbers and heterogeneous pathophysiological characteristics, which would have limited statistical power and comparability.

### 2.6. Statistical Analysis

Statistical analysis was carried out in collaboration with the Institute of Biometry and Clinical Epidemiology at Charité. SPSS Statistics 25.0 (IBM Corp., Armonk, NY, USA) was used for this purpose. First, univariate descriptive statistics were performed (Table 1). Additionally, a paired *T*-Test was used to examine mean differences between the tumor and the healthy hemisphere. Further outcome analyses were conducted for tumor entities, including glioma types using the aforementioned biomarkers, and also for patients with present paresis. Thus, the separate outcome groups were as follows:
Tumor entity: glioma, metastasis, benign;Glioma type
○WHO grade: WHO grade 2, WHO grade 3, WHO grade 4;○IDH status: mutation, wildtype;○1p19q status: codeletion, no codeletion;○Glioma entity: glioblastoma, astrocytoma, oligodendroglioma;Motor status: deficit (BMRC ≤ 4/5), no deficit (BMRC = 5/5).

First, bivariate analysis was performed to compare outcome groups regarding the four nTMS parameters as well as all patient- and tumor-specific factors as possible confounders (Appendix A). For metric variables, we used the unpaired *T*-Test or ANOVA. Categorial variables were analyzed with the Chi^2^-Test. We calculated the standardized mean difference (SMD) as a standardized effects size measure for quantifying subgroup differences. The SMD is Cohen’s d in the case of comparing two groups in a continuous measure. We used the calculation of the SMD as implemented in the R package tableone, with extensions of the SMD for nominal data.

The main analysis was performed with confounder-adjusted regression models for each outcome (Figure 2). Confounder adjustment was implemented to account for potential biases arising from inter-individual differences that could influence motor excitability measurements independently of tumor entity or motor deficit. Listwise exclusion was applied for cases with missing data to maintain consistency in regression models, and variance inflation factors (VIFs) were monitored to assess potential multicollinearity between predictors. The separate analysis models for each outcome used one single nTMS parameter (e.g., RMT sick hemisphere) together with all patient- and tumor-specific characteristics as independent variables. The strength and significance of associations for each nTMS-parameter with the outcome groups was evaluated using odds ratios (ORs) with 95% confidence intervals (CIs). Additionally, standardized mean differences (SMDs) were used to quantify effect sizes, ensuring robust comparisons across outcome groups. A two-sided significance level of α = 0.05 was used. However, no adjustment for multiple testing was applied in this exploratory analysis, and *p*-values have to be interpreted cautiously. The results are presented and visualized using forest plots to compare tumor entities (Figure 3), glioma types (Figure 4) and motor status (Figure 5).

## 3. Results

### 3.1. Population Characteristics

Our study population comprised 800 motor-eloquent-brain tumor patients who underwent motor mapping between October 2007 and March 2021 (Table 1). Motor-eloquent tumor location included M1 infiltration in 35% and a mean TTD of 6.7 mm. Motor deficits were present in 34% of the patients. The distribution of tumor entities was 58% gliomas, 24% metastases and 18% benign entities. We excluded 11 Lymphoma and 7 unclassified entities for tumor entity analysis because they were too few as an independent group and did not fit to our analyzed groups. Glioma entities comprised 54% IDH-wildtype glioblastomas, 30% IDH-mutated astrocytomas, and 17% IDH-mutated and 1p19q-codeleted oligodendrogliomas. Due to unclear neuropathological entity classification, 45 gliomas had to be excluded from the analysis of glioma types.

RMT_Sick_ (98 ± 27 V/m) and RMT_Healthy_ (97 ± 22 V/m) showed no substantial difference. The mean RMT_Ratio_ was 104 ± 22%, with 60% having a pathological RMT_Ratio_ (<90% or >110%). Area_Sick_ (306 ± 222 mm^2^) and Area_Healthy_ (307 ± 238 mm^2^) did not show a substantial difference between the hemispheres. However, Amplitude_Sick_ (597 ± 591 µV) was lower compared to Amplitude_Healthy_ (773 ± 761 µV, *p* < 0.001). Latency_Sick_ (23.5 ± 2.3 ms) and Latency_Healthy_ (23.5 ± 1.9 ms) showed no substantial interhemispheric difference.

### 3.2. Tumor Entity Analysis

The bivariate analyses of tumor entities with patient-and tumor-specific confounders and nTMS parameters can be found in Appendix A. The following results of the multiple confounder-adjusted regression analyses are shown in Figure 3.

Gliomas showed more frequently a pathologic RMT_Ratio_ compared to benign entities (OR 1.76, 95%CI: 1.06–2.89, *p* = 0.030). Benign entities exhibited shorter Latency_Sick_ compared to gliomas (OR 1.16, 95%CI: 1.01–1.33, *p* = 0.047) and shorter Latency_Healthy_ compared to metastases (OR 1.31, 95%CI: 1.03–1.67, *p* = 0.029). No substantial differences were observed for metastases compared to other entities in the tumor hemisphere.

### 3.3. Glioma Type Analysis

The bivariate analyses of glioma types with patient-and tumor-specific confounders and nTMS parameters can be found in Appendix A. The following results of the multiple confounder-adjusted regression analyses are shown in Figure 4.

Pathological RMT_Ratio_ was less frequent in the presence of IDH mutation (OR 0.43, 95%CI: 0.23–0.79, *p* = 0.006), which applied to both IDH-mutated oligodendrogliomas (OR 0.43, 95%CI: 0.20–0.93, *p* = 0.031) and IDH-mutated astrocytomas (OR 0.43, 95%CI: 0.20–0.91, *p* = 0.027) compared to IHD-wildtype glioblastomas. There was no substantial association between pathological RMT_Ratio_ and the WHO grade (OR 1.61, 95%CI: 0.96–2.71, *p* = 0.074). Lower WHO-graded gliomas exhibited a larger motor Area_sick_ (OR 0.87, 95%CI: 0.78–0.97, *p* = 0.019). Enlarged Area_Healthy_ was present in oligodendrogliomas compared to glioblastomas (OR 1.18, 95%CI: 1.01–1.39, *p* = 0.041). No substantial differences of nTMS parameters were observed for the 1p19q status.

### 3.4. Motor Status Analysis

The bivariate analyses of the clinical motor status with patient-and tumor-specific confounders and nTMS parameters can be found in Appendix A. The following results of the multiple confounder-adjusted regression analyses are shown in Figure 5. Patients with motor deficits had a higher RMT_Sick_ (OR 1.12, 95%CI: 1.05–1.21, *p* = 0.001) and RMT_Ratio_ (OR 1.17, 95%CI: 1.08–1.26, *p* ≤ 0.001). Additionally, the MEPs exhibited reduced Amplitude_Sick_ (*p* = 0.019) and prolonged Latency_Sick_ (OR 1.12, 95%CI: 1.02–1.24, *p* = 0.025) in patients with paresis.

## 4. Discussion

This study is the largest investigation of nTMS motor mappings in patients with motor-eloquent brain tumors. We assessed motor excitability neuroplasticity profiles for specific tumor entities and the clinical motor status with the help of quantitative nTMS parameters (RMT, cortical motor area, MEP amplitude and MEP latency). These parameters were chosen because they represent key aspects of motor system function: RMT reflects corticospinal excitability, the cortical motor area indicates the spatial representation of motor function, MEP amplitude measures synaptic transmission efficiency and MEP latency assesses conduction speed within the corticospinal tract. Together, these metrics provide a comprehensive evaluation of motor function alterations in brain tumor patients. The large study population made it possible to carry out analyses with adjustment for patient- and tumor-specific factors and therefore minimize effect distortion. Results in our multiple-adjusted analyses differed from those in the bivariate analyses, which supports the relevance of confounder adjustment.

### 4.1. Tumor Entities

#### 4.1.1. Glioma

Gliomas were associated with prolonged latency in the sick hemisphere compared to benign entities. A common spreading mechanism of gliomas is the infiltration of white matter fibers [46]. Several studies on motor-eloquent glioma were able to measure a reduced structural integrity of the CST fibers by parameters such as fractional anisotropy using DTI [16,47]. The integrity of the CST is also correlated with motor conduction velocity [48,49]. Therefore, we assume that prolonged latency in the sick hemisphere is a glioma-specific consequence of microstructural CST infiltration compared to non-infiltrative growing benign entities. Since we adjusted the analysis also for the TTD, this is an entity-specific effect independent of the absolute spatial distance to the CST.

Since we observed no substantial differences in gliomas compared to other entities for absolute RMT values on both hemispheres, as well as for the absolute RMT ratio, the finding of more pathological RMT ratios in gliomas compared to benign entities represents probably a bidirectional disbalanced interhemispheric excitability without clearly lateralized excitability effects. This can be possibly explained by bidirectional alterations in peritumoral motoneurons due to interactions in the glioma–brain interface. On the one hand, gliomas induce neuronal hyperexcitability by electrical and synaptic integrations in cortical networks due to various mechanisms, which contribute also to glioma-associated seizures [50,51,52]. On the other hand, many studies have demonstrated that neuronal activity promotes glioma growth [53,54,55]. Tumor progression in motor-eloquent areas increases the risk of motor decompensation, which can lead to functional hypoexcitability measured by nTMS [22,28,38,39]. Finally, this results in a vicious circle with bidirectional excitability changes in the brain–glioma interface. Additionally, the RMT ratio can also be seen as a parameter of interhemispheric balance and connectivity in the motor network. Interhemispheric balance can be altered due to neuroplastic compensation or tumor-associated decompensation, mediated by transcallosal mechanisms of inhibition and disinhibition [56,57,58]. In this context of tumor-associated brain reorganization, it is assumed that entity-specific lesion kinetics has a great impact [24,58,59]. Studies on brain tumor patients and also stroke patients show that reduced interhemispheric connectivity of the motor network measured by functional MRI is associated with motor deficits [60,61]. One study also observed pathologic RMT ratios for patients with new postoperative motor deficit after tumor resection [15]. Normalized interhemispheric connectivity, in turn, was related with a better motor recovery in stroke patients [62]. Considering the entity-specific differences in brain reorganization due to interhemispheric balance and connectivity mechanisms, we propose that these factors may account for the increased frequency of pathological RMT ratios in gliomas compared to benign entities.

#### 4.1.2. Glioma Types

In our analysis of glioma types, we observed differences between IDH-wildtype glioblastomas and IDH-mutated gliomas. IDH-wildtype glioblastomas more frequently exhibited a pathologic RMT ratio, a finding that was reproducible in comparison to IDH-mutated oligodendrogliomas and IDH-mutated astrocytomas. There are several conflicting studies on RMT alterations in glioma types, mainly investigating the WHO grade. Higher WHO-graded gliomas were related with higher absolute RMT [22,28,32], lower absolute RMT [31,39] or even no RMT differences [26,38]. Lavrador et al. were the first to evidence bidirectional pathological RMT ratios as part of the cortical excitability score in higher WHO grades and also in IDH-wildtype [31,32,33]. However, in their analysis with mutual adjustment, this correlation was robust only for the WHO grading. Since WHO grading and IDH status overlap each other very closely, effect estimation can be distorted due to perfect prediction between these two variables when integrated in the same analysis model. Our separated analyses indicate that a more frequent pathological RMT ratio is connected rather to IDH status (*p* = 0.006) than to WHO grading (*p* = 0.074). We suspect that this is also caused by different mechanisms in the brain–tumor interface, depending on the IDH. Peritumoral hyperexcitability in gliomas [50,51,52] is independent of the IDH status, but is caused by the increase in various oncometabolites [63,64,65]. Due to different energy metabolism, the tumor proliferation in IDH-wildtype is much faster [66]. Investigation of MRI-based invasion patterns by Baldock et al. revealed a local-aggressive growth for IDH-wildtype and diffuse spreading for IDH-mutation [67]. They suspected a strong relation between this different growth behavior and adaptability or reorganization of the brain. Indeed IDH-wildtype gliomas present more often with focal neurological deficits than IDH-mutated gliomas, which was also the case in our cohort. Therefore, more disruption and less adaptation of eloquent motor areas despite the same eloquent tumor location is more likely to occur in IDH-wildtype, which can lead to decompensation of motor excitability. In combination with the presence of tumor-induced hyperexcitability, this results in bidirectional excitability changes, which we observed more frequently in IDH-wildtype. This interhemispheric disbalance is also supported by two studies that found reduced interhemispheric connectivity in the motor network for IDH-wildtype compared to IDH-mutation [68,69].

Two findings of our glioma subtype analyses included changes in cortical motor area size. Lower WHO-graded gliomas showed a larger motor area on the tumor hemisphere, while oligodendrogliomas exhibited larger motor areas on the healthy hemisphere compared to glioblastomas. We suspect that the enlargement of the motor area is a sign of tumor-associated neuroplastic cortical reorganization, as this mechanism has also been proposed in many nTMS studies that found shifts of motor areas to different regions of the brain [19,21,23,70,71]. Hierarchy and mechanisms of neuroplasticity of eloquent brain functions are well described [58,59,72,73,74]. Neuroplastic capacity depends largely on entity-specific growth behavior [24,58,59]. One mechanism is ipsilesional recruitment of M1-adjacent frontoparietal brain areas, especially premotor cortex regions. This may account for our observation of larger motor areas in the tumor-affected hemisphere for low-grade gliomas and underscores the influence of biological tumor aggressiveness on neuroplasticity. Another neuroplastic mechanism that occurs probably later in the hierarchy is the recruitment of contralesional underused or inhibited homolog motor areas, which is assumed to be mediated by tumor-induced transcallosal disinhibition. Our results demonstrate that motor area enlargement on the contralesional hemisphere is more likely to be found in slower growing oligodendrogliomas compared to glioblastomas, which underlines the impact of entity-specific lesion kinetics on neuroplasticity.

#### 4.1.3. Metastasis

Brain metastasis showed no substantial alterations of nTMS parameters in the tumor hemisphere compared to both other entities. Previous nTMS studies compared metastases to gliomas and did not find specific differences [28,38,39]. One study from Eibl et al. [26] found more frequent pathological RMT ratios in metastases but did not adjust for the motor status and edema volume. Research on invasion patterns in the brain–metastasis interface showed very heterogenous results from local demarcation to aggressive infiltration [75,76]. This was related rather with specific molecular characteristics than with tumor primaries. Unfortunately, subgroups of various metastases have not yet been investigated by nTMS. Interestingly, the only substantial nTMS parameters alteration for brain metastases were prolonged latencies on the healthy hemisphere compared to benign entities, which indicates impaired motor conductivity. It is known that metastases grow multifocally in the entire brain in up to 85% [77], which is why we adjusted our analysis for tumor foci. We ruled out the motor-eloquent location of other tumor foci, even if contralaterally located, but did not further investigate those foci. Therefore, we can only speculate whether this may be caused by the mass effect on non-eloquent suspected foci, the presence of other radiological non-visible foci or neurotoxic effects due to previous radiotherapy or chemotherapy. However, a recent experimental animal study with unilateral implanted metastases also found bihemispheric impairment of electrophysiological signals. Summarized, we found no evidence of different nTMS motor profiles on the tumor hemisphere for metastases versus gliomas and benign entities, but rather unspecific phenomena for motor impairment on the healthy hemisphere.

#### 4.1.4. Benign Entites

Benign entities differ in terms of (de)compensatory changes in the nTMS motor profile compared to malignant entities, despite having the same motor-eloquent localization. They showed shorter latencies than gliomas in the tumor hemisphere and shorter latencies than metastases in the healthy hemisphere. Therefore, the latency seems to be a sensitive nTMS parameter, to differentiate benign from malignant entities. Additionally, the RMT ratio was more frequent physiologically balanced than in gliomas. These findings underline the biological benignity in this group an indicate preservation of normal integrity and functionality in the motor network.

### 4.2. Motor Status

Patients with clinical motor deficits had a higher RMT, lower amplitude and longer latency in the tumor hemisphere. Absolute values of the interhemispheric RMT ratio were also higher in patients with motor deficits compared to those without deficits. Picht et al. suspected changes in these three parameters are tumor-mediated signs of disrupted integrity and excitability in the motor system [27]. Numerous nTMS studies observed asymmetric hypoexcitability in the brain tumor hemisphere [22,28,38,39]. In contrast to the observed bidirectional excitability changes in gliomas and specifically in IDH-wildtype, motor deficits lead to clearly unidirectional hypoexcitability that is not associated per se with pathological RMT ratios. Since motor impairment in stroke patients was related to higher transcallosal inhibition from the healthy to the sick hemisphere [56], we assume similar mechanisms can lead to hypoexcitability in brain tumor patients. A higher RMT was also associated with postoperative motor worsening after motor-eloquent brain tumor resection [16,18]. Reduced amplitude and longer latency have not yet been explicitly associated with motor deficits in nTMS studies; however, they are commonly observed in motor-eloquent brain tumor patients [9,22,28,37,47] and are associated with poor postoperative motor outcomes when present during intraoperative monitoring [78,79,80]. Interestingly, the amplitude was the only nTMS parameter that showed a substantial mean difference between both hemispheres in our study although only 34% of patients had motor deficits. This implies that amplitude values from both hemispheres, such as through the calculation of an interhemispheric ratio, may help identify patients at risk for motor decompensation, even in the absence of a manifest paresis. A lower amplitude in the affected hemisphere may indicate impaired corticospinal excitability, potentially signifying a reduced capacity for functional compensation postoperatively. Thus, incorporating amplitude measurements into preoperative risk stratification models may improve patient selection for extended monitoring and tailored neuromodulatory interventions to preserve motor function.

### 4.3. Clinical Relevance

Our findings suggest that entity-specific and motor-status-specific excitability changes reflect different compensatory and decompensatory mechanisms in the motor system, indicating different extents of adaptive plasticity or vulnerability to functional deterioration. These patterns align with previous research on tumor-induced neuroplasticity and reinforce the role of tumor biology in shaping motor network reorganization [24,58,59]. Recognizing these patterns preoperatively allows for a patient- and tumor-specific surgical strategy, balancing tumor resection aggressiveness with preservation of brain functionality. Integrating nTMS-based excitability markers into preoperative planning can help stratify surgical risks, refine tumor–border delineation and optimize resection margins to improve patients outcomes, as already shown in some nTMS studies [15,16,17,18]. Risk stratification models have so far been developed on cohorts with heterogeneous brain tumor entities. Based on the entity-specific changes in the nTMS profile, these risk stratification models could be further optimized and adapted for different tumor entities.

Beyond surgical planning, these insights could also inform neuromodulatory interventions aimed at enhancing motor recovery and functional compensation. Preoperative nTMS-based stimulation paradigms or rehabilitation strategies tailored to excitability profiles should be explored to promote neuroplasticity and mitigate postoperative deficits. Future studies should explore how stimulation-induced plasticity mechanisms could be utilized to enhance motor system resilience in high-risk patients.

### 4.4. Limitations

Due to the retrospective study design and the evaluation of a large study population over a long time period, we were unable to obtain complete data for all parameters. However, listwise exclusion of cases with missing data enabled analyses with full data sets. Regarding nTMS motor mapping, some limiting factors must also be considered. At first, inter-examiner variability influenced the nTMS mapping results, which is why a standardized mapping protocol was applied to reduce this effect. Additionally, we only evaluated the hand muscle FDI, which is most frequently used in nTMS studies. We cannot verify our results for other extremities and muscles, although we assume that comparable relationships exist. Furthermore, nTMS motor mapping was performed only on a single time point. Therefore, our analyzed quantitative nTMS parameters did not enable precise temporospatial evaluation of neuroplasticity, and such effects can only be suspected in this study. The importance of various patient- and tumor-specific factors that contribute to nTMS parameter variability has also been explained in this study. In this context, our study is the first to include so many relevant confounders in the analysis models to minimize effect distortion. Certainly, motor functionality can be affected by many more factors, which we did not include (e.g., previous radiotherapy or chemotherapy, location or metrics of other non-eloquent tumor foci, occurrence of seizures). Selection and inclusion of confounders in statistical analysis must be well thought out to ensure independence and prevent multicollinearity of those predictors. Since nearly all patients with epileptic seizures were also receiving AED and vice versa, including both predictors in the same model is not feasible due to collinearity concerns, which could distort effect estimation. This could also explain why two nTMS studies found surprisingly lower RMTs in patients taking levetiracetam when performing backwards regression with inclusion of both variables, among others [38,39]. Although this study did not include data from children with brain tumors, there are several studies that demonstrate the successful deployment of nTMS mapping in children [81,82].

As a last limiting factor, the grouping of different non-malignant entities as benign entities can be discussed due to heterogenous subentities. On the other hand, this enabled stable statistical analysis and comparison of nTMS motor excitability between malignant and benign entities, which has not been investigated before. However, subtypes of heterogenous groups like benign entities and also metastases need to be examined more specifically with nTMS to get a better understanding of different brain tumor interactions in the motor network.

## 5. Conclusions

This is the largest published study to date on nTMS motor mappings in brain tumor patients, in which we examined motor excitability profiles using a confounder-adjusted regression analysis. We identified balanced interhemispheric RMT ratios, motor area enlargements and shorter latency times as features of benign entities and biologically less aggressive lower-graded IDH-mutated gliomas. These nTMS-based alterations of motor excitability highlight various compensatory neuroplastic mechanisms. In contrast, the opposite excitability alterations of aggressively growing IDH-wildtype glioblastomas indicate greater vulnerability and reduced plasticity of the motor system. However, decompensatory changes in nTMS parameters due to clinical motor deficits differentiate from entity-specific patterns.

Therefore, our findings in motor-eloquent brain tumor patients provide fundamental insights into entity-specific brain–tumor interactions in the motor system. This knowledge should be used to optimize individual risk assessment before tumor resection in order to preserve motor function of patients.

## Figures and Tables

**Figure 1 cancers-17-00935-f001:**
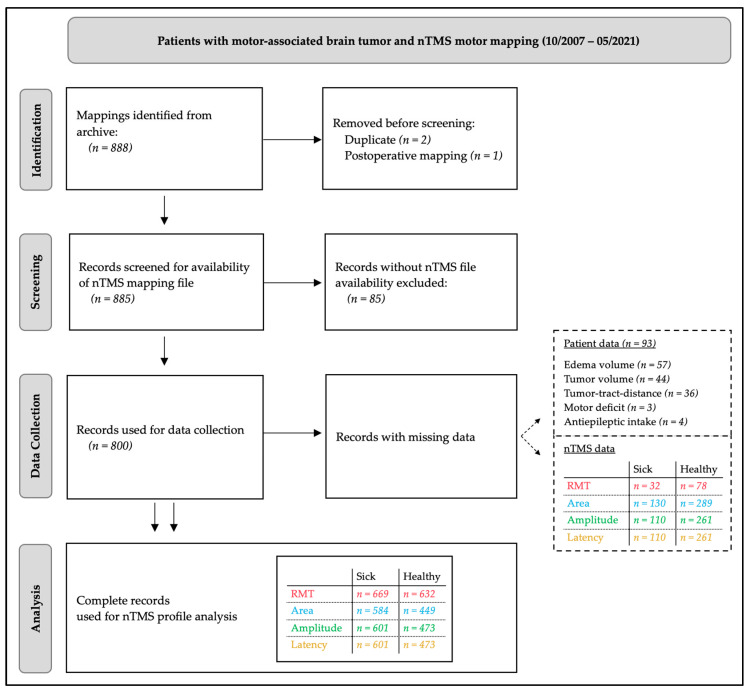
STROBE flowchart for selection process of the study population.

**Figure 2 cancers-17-00935-f002:**
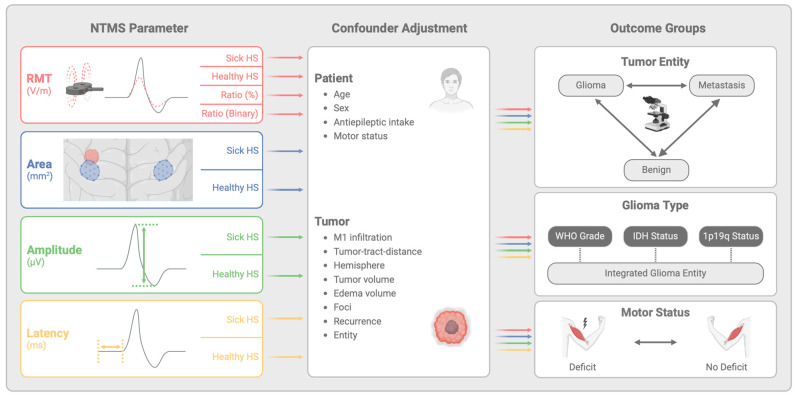
Multiple-adjusted regression analysis concept as flowchart. NTMS profile analysis was performed by analyzing each single nTMS parameter separately, with adjustment for all patient and tumor confounders for changes in our outcome groups. HS—hemisphere; M1—motor cortex; WHO—world health organization; IDH—isocitrate deyhdrogenase. Created with BioRender.com.

**Figure 3 cancers-17-00935-f003:**
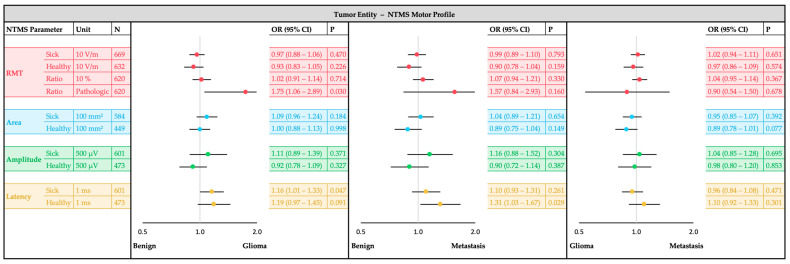
NTMS motor profile comparison for tumor entities (glioma, metastasis, benign). Forest plots of the multiple logistic regression analysis showing the odds ratio (OR) and the 95% confidence interval (CI) of each nTMS parameter. Each single nTMS parameter was analyzed separately with adjustment for sex, age, antiepileptic intake, motor status, motor cortex (M1) infiltration, tumor-tract-distance (TTD), tumor hemisphere, tumor volume, edema volume, tumor foci and tumor recurrence.

**Figure 4 cancers-17-00935-f004:**
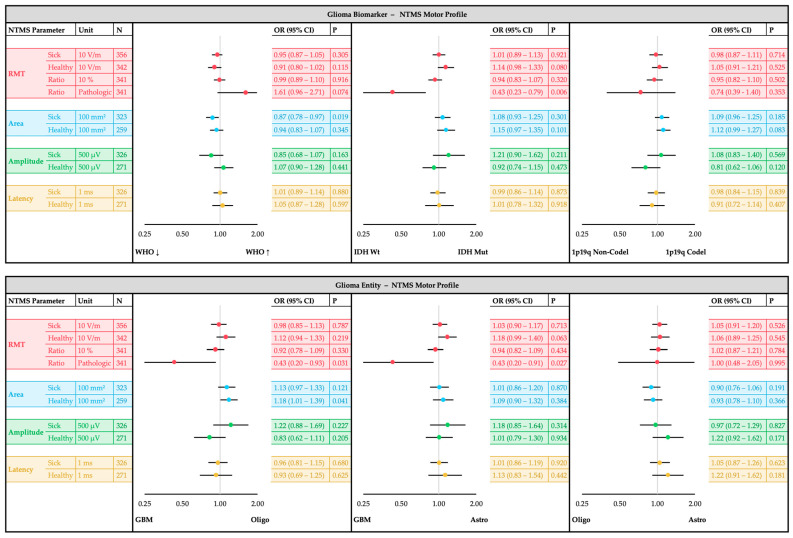
NTMS motor profile comparison for glioma types. Top: Diagnostic glioma biomarker (WHO grading, IDH status, 1p19q status). Below: Integrated glioma entity (oligodendroglioma, astrocytoma, glioblastoma). Forest plots of the multiple logistic regression analysis showing the odds ratio (OR) and the 95% confidence interval (CI) of each nTMS parameter. Each single nTMS parameter was analyzed separately with adjustment for sex, age, antiepileptic medication, motor status, motor cortex (M1) infiltration, tumor-tract-distance (TTD), tumor hemisphere, tumor volume, edema volume, tumor foci and tumor recurrence. WHO—world health organization; IDH wt/mut—isocitrate dehydrogenase wildtype/mutation; GBM—glioblastoma; Oligo—oligodendroglioma; Astro—astrocytoma.

**Figure 5 cancers-17-00935-f005:**
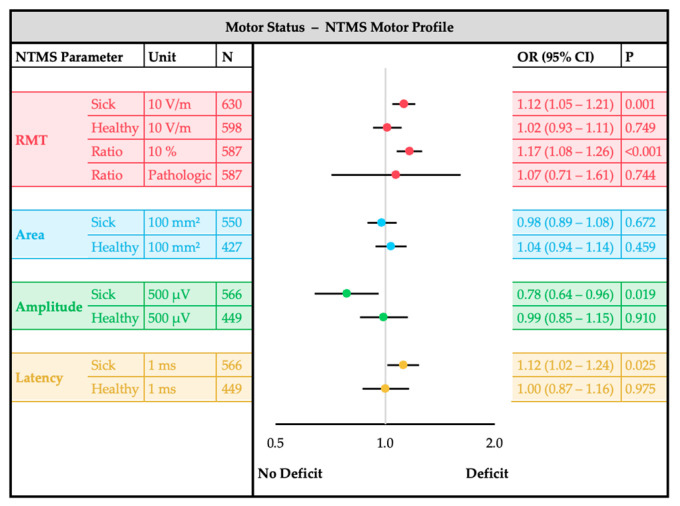
NTMS motor profile comparison for motor status (deficit or no deficit). Forest plots of the multiple logistic regression analysis showing the odds ratio (OR) and the 95% confidence interval (CI) of each nTMS parameter. Each single nTMS parameter was analyzed separately with adjustment for sex, age, antiepileptic intake, motor cortex (M1) infiltration, tumor tract-distance (TTD), tumor hemisphere, tumor volume, edema volume, tumor foci, tumor recurrence and tumor entity.

**Table 1 cancers-17-00935-t001:** Population Characteristics.

Variable		*n* (%)	Mean (SD)	*p* ^1^	SMD ^2^ (95%CI)
**Patient Charateristics**					
Female		376 (47.0%)			
Age (y)			52.9 (15.5)		
Antiepileptic Medication		393 (49.4%)			
Motor Deficit (BMRC ≤ 4)		267 (33.5%)			
**Tumor Location and Morphology**					
Motor Location	M1-TMS-Infiltration	276 (34.5%)			
	TTD (mm)		6.7 (6.8)		
Dominant Hemisphere		358 (45%)			
Tumor Volume (mL)			22.3 (25.3)		
Edema Volume (mL)			40.8 (45.4)		
Multifocal (≥2 Foci)		180 (23%)			
Tumor Recurrence		187 (23%)			
**Neuropathology**					
Tumor Entity ^3^	Glioma	456 (58%)			
	Metastasis ^4^	185 (24%)			
	Benign ^5^	141 (18%)			
Glioma Type ^6^					
WHO Grade	WHO 2	54 (13%)			
	WHO 3	106 (26%)			
	WHO 4	251 (61%)			
IDH Mutation		190 (46%)			
1p19q Codeletion		68 (17%)			
Glioma Entity	Oligodendroglioma	68 (17%)			
	Astrocytoma	122 (30%)			
	Glioblastoma	221 (54%)			
**NTMS Parameter**					
RMT	Sick (V/m) (32 missings)		98 (27)	0.954	−0.05 (−0.15 to 0.05)
	Healthy (V/m) (78 missings)		97 (22)		
	Ratio (%)		104 (28)		
	Ratio (Pathologic)	423 (60%)			
Area	Sick (mm^2^) (130 missings)		306 (222)	0.207	0.01 (−0.11 to 0.12)
	Healthy (mm^2^) (289 missings)		307 (238)		
Amplitude	Sick (µV) (110 missings)		597 (591)	<0.001	0.26 (0.15 to 0.37)
	Healthy (µV) (261 missings)		773 (761)		
Latency	Sick (ms) (110 missings)		23.5 (2.3)	0.199	0.04 (−0.07 to 0.15)
	Healthy (ms) (261 missings)		23.5 (1.9)		

BMRC—British Medical Research Council. M1—Motor Cortex. TTD—Tumor-Tract-Distance. WHO—World Health Organization. IDH—Isocitrate Dehydrogenase. RMT—Resting Motor Threshold. ^1^
*p*-values assess mean differences between both hemispheres of each nTMS parameter. ^2^ Standardized mean difference, standardized effect sizes measure ^3^ 18 other tumor entities (excluded from analysis): 11 lymphoma and 7 unclassified; ^4^ 185 metastases: 98 bronchial, 28 mamma, 21 malignant melanoma, 16 urogenital, 12 gastrointestinal and 10 other; ^5^ 141 benign entities: 72 vascular malformations, 37 meningioma, 12 gliosis/brain tissue, 12 encephalitis and 8 other primary brain tumors WHO grade 1; ^6^ 45 unclassified glioma subtypes (excluded from glioma type analyses).

## Data Availability

The raw data supporting the conclusions of this article will be made available by the authors on request.

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
