# Peer review of "Analysis of Neuronal Excitability Profiles for Motor-Eloquent Brain Tumor Entities Using nTMS in 800 Patients"

_cancers, 2025, doi:10.3390/cancers17060935_

Round 1

Reviewer 1 Report

Comments and Suggestions for Authors

This manuscript reports a large, well conducted study discovering that different brain cancers exhibit different rTMS motor excitability traits. Importantly, it distinguishes between less aggressive and more aggressive entities. Of interest, they found evidence that amplitude values might predict later paresis. Please say a little more about how this knowledge could help to assess surgical risk. There are some typos. In line 374, mediate should be mediated. In line 393, where should be were. In line 418, changes should be followed by in.

Reviewer 2 Report

Comments and Suggestions for Authors

This paper addresses motor excitability profiles in patients with motor-eloquent brain tumors using navigated transcranial magnetic stimulation (nTMS). This work can be extended with practical considerations with preoperative planning and assessing the functional integrity of motor pathways affected by tumors.

The following changes are suggested:

  1. In the introduction, clearly state the research questions you are gong to address in this study.
  2. The method you followed, STROBE, is a primarily a reporting guideline. So, how do you assess the quality of the methodology.

  1. By following STROBE, are there any factors omitted, that affect the sequence of events?

  1. The organization of the paper, sections, sub sections, sub-sub sections, bullet points should be refined and improve the structure.

  1. Justify the selected nTMS approach. How does nTMS compare to other methods in assessing motor excitability in brain tumor patients.

  1. What is the possibility of using nTMS with DES and fMRI. What will be the complications and processing time requirements in such an integration.

  1. How do you assess the generatlizability of this approach. Because nTMS is less effective in pediatric patients due to immature axonal myelination, behavioral issues, and lower neuronal response rates. So, how do you address the patients with young age and neurodevelopmental impairments in mapping.

  1. Under the description of different analysis (eg. 2.5, 2.6), it would be better to state some scientific/ technical background of those analyses as well.

  1. It would be better to improve the clarity of the write-up. For example, in line 341, you have mentioned that , “using four quantitative nTMS parameters…..” , State what are those 4, and justify for considering those .

  1. Under results analysis, better if you could include some graph visualizations, for the clear understanding of the results analysis.

  1. It would be better to discuss how the proposed approach can help in surgical planning

  1. Discuss how brain tumors affect neuronal excitability.

Reviewer 3 Report

Comments and Suggestions for Authors

This study examines motor excitability profiles in 800 patients with motor-eloquent brain tumors through navigated transcranial magnetic stimulation (nTMS). It discovers distinct alterations in nTMS parameters across various tumor types, especially gliomas, with IDH-wildtype gliomas exhibiting more pathological resting motor threshold (RMT) ratios, while lower WHO-graded gliomas feature larger motor areas. Benign tumors showed shorter latencies and balanced RMT ratios, indicating preserved motor network integrity. Motor deficits were linked to higher RMT, decreased amplitude, and increased latency. These findings underscore tumor-specific neuroplasticity and the vulnerability of the motor system, informing surgical risk evaluation and enhancing functional outcomes. The results offer crucial insights into the motor excitability profiles specific to different tumors, emphasizing the ability of nTMS to improve preoperative risk assessment and surgical results by revealing neuroplasticity patterns and vulnerabilities in brain tumor patients. I believe the paper merits publication in Cancers with minor corrections.

Comments for authors

Comment 1. The word “Motor” in the title appeared twice, consider revising it.

Comment 2. Provide details on the nTMS mapping protocol, including stimulation parameters, coil type, and how inter-examiner variability was minimized.

Comment 3. Why were specific subgroups (lymphoma, unclassified tumors) excluded? Explain the reason in the manuscript.

Comment 4. Clarify the criteria for motor deficits (BMRC ≤4/5) and discuss potential biases in motor status assessment.

Comment 5. Explain how tumor volume and edema volume were assessed and their potential impact on nTMS parameters in the discussion section.

Comment 6. The manuscript contains several grammatical and language errors that occasionally hinder readability and clarity. Improving English will help it meet high academic writing standards. Review and correct the grammatical errors in the revised version.

Comments on the Quality of English Language

The manuscript contains several grammatical and language errors that occasionally hinder readability and clarity. Improving English will help it meet high academic writing standards. Review and correct the grammatical errors in the revised version.

Reviewer 4 Report

Comments and Suggestions for Authors

The study examines motor excitability profiles in patients with motor-eloquent brain tumors using navigated transcranial magnetic stimulation (nTMS). It analyzes data from 800 patients to determine whether specific tumor entities, including gliomas, metastases, and benign tumors, exhibit characteristic alterations in nTMS parameters such as resting motor threshold (RMT), motor area size, amplitude, and latency. The findings suggest that different tumor types influence motor excitability in distinct ways, with gliomas showing higher rates of pathological RMT ratios and benign tumors maintaining better motor integrity. The study highlights the role of tumor biology in neuroplasticity and its potential implications for surgical risk assessment.

While the large cohort and rigorous statistical analyses strengthen the study, its retrospective design and incomplete data for some patients present limitations. The grouping of heterogeneous benign entities and the lack of longitudinal nTMS assessments restrict the ability to track neuroplastic changes over time. 

Some suggestions: 

  • The quality of figures should be improved to meet current scientific standards. Higher resolution, clearer labeling, and standardized formatting (e.g., consistent colors and axis labeling) would enhance readability, particularly for forest plots.

  • Some references are outdated, and more recent studies on neuroplasticity, glioma biology, and nTMS applications should be incorporated to strengthen the discussion and contextualize findings.

  • The integration of results with existing literature could be improved. A deeper exploration of how tumor-specific excitability changes relate to functional outcomes would strengthen the clinical relevance.

  • A clearer explanation of how missing data were handled and how potential confounders (e.g., prior treatments, epilepsy) were addressed would improve transparency.

  • Further discussion on how these findings could inform neuromodulatory interventions or preoperative decision-making would add to the study’s impact.
